# Climatic Variabilities Control the Solute Dynamics of Monsoon Karstic River: Approaches from *C-Q* Relationship, Isotopes, and Model Analysis in the Liujiang River

**Jing Liu [1],\*, Hu Ding [2]**  **, Min Xiao [3],\*, Zhu-Yan Xu [1], Yuan Wei [1], Zhi-Hua Su [1], Lei Zhao [1], Jiao-Ting Peng [1], Heng Wang [4] and Xiao-Dan Wang [5]**

1   School of Management Science, Guizhou University of Finance and Economic, Guiyang 550025, China;
    199501016@mail.gufe.edu.cn (Z.-Y.X.); 200801075@mail.gufe.edu.cn (Y.W.);
    201201063@mail.gufe.edu.cn (Z.-H.S.); 201901040@mail.gufe.edu.cn (L.Z.);
    200801038@mail.gufe.edu.cn (J.-T.P.)
2   State key laboratory of Environmental Geochemistry, Institute of Geochemistry, Chinese Academy of
    Sciences, Guiyang 550081, China; dinghu@vip.skleg.cn
3   Tianjin Key Laboratory of Water Resources and environment, Tianjin Normal University,
    Tianjin 300387, China
4   School of Public Managemerent, Guizhou University of Finance and Economics, Guiyang 550025, China;
    201401099@mail.gufe.edu.cn
5   School of Eco-Environment Engineering, Guizhou Minzu University, Guiyang 550025, China;
    wangxiaodan@gzmu.edu.cn
*   Correspondence: 201701057@mail.gufe.edu.cn (J.L.); xiaomin@vip.skleg.cn (M.X.)

**Abstract:** The dynamics of riverine solutes' contents and sources reflect geological, ecological, and climatic information of the draining basin. This study investigated the influence of climatic variability on solute dynamics by the high-frequency hydrogeochemical monitory in the Liujiang River draining karst terrain of Guangxi Province, SW (Southwestern) China. In the study river, the content-discharge (*C-Q*) patterns of riverine solutes indicate that the majority of riverine solutes show similar dilution and near chemostatic behaviors responding to increasing discharge, especially geogenic solutes (such as weathering products from carbonate, silicate, and sulfide oxidation), whereas exogenous solutes (such as atmospheric input to riverine sulfate) and biological solutes (such as soil $CO_2$) show higher contents with increasing discharge. Besides, the biological carbon is the main driver of the chemostatic behaviors of total dissolved inorganic carbon (DIC). The forward model results show that carbonate weathering dominates the water chemistry, and the weathering rates are intensified during high flow period due to additional inputs of weathering agents, i.e., the biologic carbonic acid from dissolution of soil $CO_2$, indicated by $\delta^{13}C_{DIC}$. In addition, there exists the strong capacity of $CO_2$ consumption that is heavily dependent on climatic variables such as precipitation and air temperature in this study river. Our study highlights the impact of climatic variability on solutes dynamics and chemical weathering and thus must be better addressed in C models under future climate change scenarios.

**Keywords:** solute–discharge relationship; stable isotopes; chemical weathering; $CO_2$ consumption; climatic variability

## 1. Introduction

$CO_2$ consumption during rock chemical weathering by reaction with carbonic and other strong acids (such as sulfuric and nitric acids) are part of the important biogeochemical cycle of carbon and therefore act on regulating the climate on Earth, even on shorter timescales [1–3]. Numerous studies have focused on chemical weathering and $CO_2$ consumption in carbonate-dominated catchments to understand local and even global carbon cycles [3,4]. Current estimation of global $CO_2$ consumption by rock weathering varies from 0.1 to 0.44 Gt C $a^{-1}$ [4,5]. These estimates have some uncertainty, largely due to the spatial variations (such as lithology, soil development, vegetation, precipitation, temperature, anthropogenic activity, etc.), which are inevitable on the continent [6]. Concerning the spatial variations in weathering flux and global $CO_2$ consumption flux, for example, a long-term monitoring system called the Hydrological Benchmark Network (HBN) was established by the United States Geological Survey (USGS) to assess and quantify the human influence on 59 study sites across the U.S. [7]. Subsequently, a temporal rather than a spatial approach may be possible to obtain a stronger correlation between weathering flux and climate.

Solutes' *C-Q* relationships in various catchments have been explored in the past decades [2,8–17] and can represent the integration of hydrological and biogeochemical responses of catchments for understanding riverine solute source, transport, and reaction [17]. The slope (*b*) of a power-law function [14] and the ratio of the *C-Q* coefficient of variation ($CV_C/CV_Q$) [18] have been proposed to evaluate the *C-Q* patterns to identify functional linkages between catchment hydrology and biogeochemistry. When $b = -1$ represents decreasing solute content with increasing discharge [14], this would support dilution behaviors where the solute mass does not increase proportionally to the increasing discharge, whereas positive *b* indicates increasing solute content with increasing discharge [19] and supports flushing behaviors. A solute is typically characterized as source-limited if it dilutes, whereas it is defined as transport-limited if it shows flushing behavior [20]. Thompson et al. [18] emphasized the importance of $CV_C/CV_Q$, which is to facilitate a more nuanced interpretation of *C-Q* relationships, particularly when $b \approx 0$. Because it is related to the "chemostatic" behavior [14] or the "biogeochemical stationarity" [21], this implies that solute content shows a negligible variability. Based on quantitative metrics that were reported by Musolff et al. [19], chemostatic behavior yielded as $-0.2 < b < 0.2$ and $CV_C/CV_Q < 0.5$. In contrast, chemodynamic behavior ($b \approx 0$, $CV_C/CV_Q > 1$) is a discharge-independent status, indicating dissolved solute content is not controlled by *Q*.

This study contributes to this line of research to investigate a high-frequency sampling survey in the Liujiang River catchment draining through the carbonate-dominated area, which features a warm subtropical climate. We focused on temporal research to (1) explore the behaviors of riverine solutes in a hydrological year by means of *C-Q* relations; (2) understand the hydrological and the biogeochemical responses of chemical weathering, $CO_2$ consumption, dissolved carbon, and sulfur dynamics in a typical karst river; (3) trace water, riverine sulfate, and dissolved inorganic carbon (DIC) sources and estimate their contributions constrained by stable isotopic tracers in the study catchment area under various climatic conditions.

## 2. Materials and Methods

### 2.1. Study Area

The Liujiang River is a first-order tributary of the Xijiang River consisting of Pearl catchment in southern China (Figure 1). It originates from the village Lang in Guizhou Province and flows through Guizhou, Guangxi and Hunan Provinces, with 72% of its drainage area in the Guangxi Province; the drainage area is 58,270 $km^2$. The main channel length is 1121 km. It is a mountainous watershed with high mountains in the north and a high elevation in the northwest, whereas, in the southern and the southeast areas, the elevations are relatively low. There are six land use types, with average coverage ratios as follows: forestland (64.9%) > cropland (18.7%) > grassland (18.1%) > urban land (14.5%) > water (0.8%) > unused land (0.02%). The Liujiang River catchment is exposed by a subtropical humid

under monsoonal climate with a mean annual precipitation of 1800 mm. It is in the center of the storm zone of the Guangxi Province with frequent storms, and 59 disastrous flooding events have been recorded in the past 400 years since 1488 [22]. There are no significant reservoirs to influence flood discharge in the study catchment. Lithologically, the Precambrian metamorphic rocks and the quaternary fluvial sediments are distributed in the whole Xijiang River catchment (Figure 1). Specifically, carbonate rocks (limestone and dolomite) and coal-bearing formations generally enriched in sulfides are widely distributed in the upper-middle reaches. Schist, gneiss and granite are exposed in the middle-lower reaches. Shale and red sandstone are distributed in the source area and are fragmentarily intercalated in the middle catchment area. Minor evaporites are scattered in the Xijiang River catchment, but a salt-bearing stratum has not been found in this area [1,23,24]. Karst topography is well-developed in the study catchment.

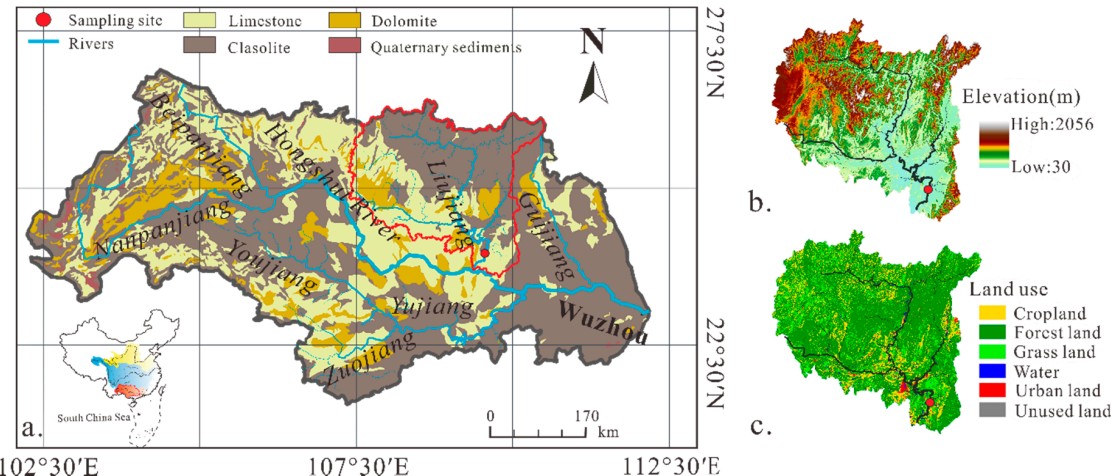

**Figure 1.** Map showing sampling location, geological background, digital elevation model (DEM), and land use types of the Liujiang River catchment.

## 2.2. Sampling and Analysis

The sampling site was located at the outlet of the Liujiang River (Figure 1), approximately 29 km away from the mainstream of the Xijiang River. River water samples for chemical and isotopic analyses were collected monthly from October 2013 to September 2014 (Table S1). Additional samples were collected in the high flow season, covering hydrological variations in this period. Water samples were collected from a boat in the middle of the river. Alkalinity was determined using 0.02 M HCl titration within 24 hours. Samples were filtered through 0.45 μM cellulose-acetate membrane paper and then were further separated into two parts; one for anions ($Cl^-$, $SO_4^{2-}$ and $NO_3^-$) determined by ionic chromatography with a precision of 5% and the other for major cations ($K^+$, $Na^+$, $Ca^{2+}$ and $Mg^{2+}$) and Si, which were acidified to pH ≤ 2 with ultra-purified $HNO_3$ and then determined by inductively coupled plasma-optical emission spectrometry (ICP-OES) with precisions better than 3%.

For the $\delta^{13}C_{DIC}$ analyses, based on the method of Li et al. [25], 15 ml aliquots of water samples were injected into vacuum glass bottles pre-filled with 2 ml 85% phosphoric acid and a magnetic stirrer bar. The samples were heated at 50 °C to extract $CO_2$ in a vacuum line and transferred cryogenically into tubes. The values of $\delta^{13}C_{DIC}$ were measured by Finnigan MAT 252 mass spectrometer and were expressed in as permil deviation with reference to a standard (VPDB), with a precision of 0.1‰. The measurement of $\delta^{13}C_{DIC}$ was conducted in the State Key Laboratory of Environmental Geochemistry, the Institute of Geochemistry, Chinese Academy of Science. Riverine sulfate was precipitated as $BaSO_4$ through adding excess $BaCl_2$ solution after the water was acidified using HCl for isotopic measurements of sulfate. Then, the precipitate was filtered, washed, and dried. The values of $\delta^{34}S_{SO4}$ and $\delta^{18}O_{SO4}$ were determined using elemental analysis-isotope ratio mass spectrometry

(EA-IRMS) and reported using δ notation relative to the Vienna Canyon Diablo Troilite (V-CDT) with precision better than 0.2‰ and the Vienna Standard Mean Ocean Water (V-SMOW) in permil with precision better than 0.5‰, respectively. The measurement of S and O isotopes of riverine sulfate was carried out at the Institute of Geographic Sciences and Natural Resources Research, Chinese Academy of Sciences. Daily water discharge data ($m^3$/s) were obtained online from the Ministry of Water Resources (http://www.hydroinfo.gov.cn/).

## 3. Results

### 3.1. Hydrochemistry

Liujiang River water is mildly alkaline, with pH value ranging from 7.5 to 8.1. Electrical conductivity (EC) value varies from 148 to 229 μS/cm, with an average of 191. Total dissolved solid (TDS = $Ca^{2+}$ + $Mg^{2+}$ + $Na^+$ + $K^+$ $HCO_3^-$ + $SO_4^{2-}$ + $Cl^-$ + $NO_3^-$ + $SiO_2$, mg/L) of river varies from 128 mg/L to 224 mg/L, with a mean value of 163 mg/L for the study river, which is higher than the world average value of 97 mg/L [16]. The total cationic charge ($TZ^+$ = $K^+$ + $Na^+$ + $Ca^{2+}$ + $Mg^{2+}$) and the total dissolved anionic charge ($TZ^-$ = $HCO_3^-$ + $Cl^-$ + $NO_3^-$ + $SO_4^{2-}$) are well balanced within all NICB (Normalized Ionic Charge Balance) (NICB = ($TZ^+$ − $TZ^-$) × 100%/($TZ^+$ + $TZ^-$)) below 5%. Similar to the rivers of Beipan and Nanpan in the upper reaches of the Xijiang River [1], $Ca^{2+}$ and $Mg^{2+}$ are dominant cations, while $HCO_3^-$ and $SO_4^{2-}$ are the dominant anions, indicating that those waters are of karstic type. The mean contents of major cations are as follows: $Ca^{2+}$ (806 μmol/L) > $Mg^{2+}$ (185 μmol/L) > $Na^+$ (111 μmol/L) > $K^+$ (28 μmol/L). The mean contents of major anions are as follows: $HCO_3^-$ (1612 μmol/L) > $SO_4^{2-}$ (125 μmol/L) > $Cl^-$ (110 μmol/L) > $NO_3^-$ (83 μmol/L). Contents of $Cl^-$, $NO_3^-$, $K^+$, and $Na^+$ are relatively low.

### 3.2. $\delta^{34}S_{SO4}$, $\delta^{18}O_{SO4}$, and $\delta^{13}C_{DIC}$ Values

In the study river, the isotopic compositions of riverine sulfate show a narrow range for $\delta^{34}S_{SO4}$ value from −0.5‰ in the high flow season to −0.1‰ in the low flow season, with a mean value of −0.4‰; for $\delta^{18}O_{SO4}$, the corresponding range is 3.9‰–10.1‰ with a mean value 6.8‰, showing a distinct temporal variation. More depleted $\delta^{18}O_{SO4}$ values are observed in the high flow season relative to those in the low flow season for the study river. The S and O isotopic compositions of sulfate in this study are in agreement with previous studies [26].

In order to calculate the partial pressure of $CO_2$ ($pCO_2$), we used the temperature dependence of thermodynamic constants [27]. The $pCO_2$ ranges from 846 μatm in the high flow season to 3999 μatm in the low flow season with a mean value of 1612 μatm in river water, two to eleven times higher than that of the atmosphere (349 μatm). The $\delta^{13}C_{DIC}$ values range from −16.2‰ in the low flow season to −8.0‰ in the high flow season, with an average value of −12.9‰, also showing a clear temporal change.

## 4. Discussion

### 4.1. Solute Content-Discharge (C-Q) Relationship

In this study river, the slopes (*b*) are negative, and $CV_C/CV_Q$ < 0.5 for weathering products such as $Ca^{2+}$, $Mg^{2+}$, $HCO_3^-$, $Na^+$, $K^+$, $SO_4^{2-}$, and $Cl^-$ signal an inverse relationship between solute content and discharge and indicate dilution, which is common for geogenic solutes, as shown in Figures 2 and 3. Geogenic solutes from carbonate weathering (such as $Ca^{2+}$, $Mg^{2+}$, and $HCO_3^-$) show near chemostatic behavior, which can be attributed to the fast kinetics of carbonate weathering processes [12,28]. $Na^+$ exhibits the strongest dilution pattern among all weathering solutes, with a more negative *b* and lower $CV_C/CV_Q$ values. $Na^+$ is mainly sourced from silicate weathering, implying that silicate weathering is more sensitive to various hydrologic conditions relative to carbonate weathering (indicated by $Ca^{2+}$). Except silicate-sourced weathering, $K^+$ is generally considered to be controlled by soil in water

flow with cation exchange [29]. Hence, K$^+$ can be considered a biological solute, especially during most storms with large amounts of soil water inputs [17]. As with K$^+$, SO$_4$$^{2-}$ and Cl$^-$ present similar dilution and chemostatic behaviors, which can be attributed to the geogenic and the exogenous sources. Exogenous sources are closely associated with soil water as well as atmospheric and anthropogenic inputs to these ions. Previous studies in this area demonstrated that sulfuric acid is also an important agent of rock weathering by using $^{87}$Sr/$^{86}$Sr and δ$^{13}$C [1,24]. The quantitative analysis of the sulfate source of this investigated river is discussed below. NO$_3$$^-$ contents are the most variable relative to discharge, implying that exogenous anthropoge nic-printed sources (such as chemical fertilizer, soil organic nitrogen, manure, sewage waste, and atmospheric input) and relevant biogeochemical processes (assimilation, nitrification, denitrification) counteract the dilution effects. However, Si shows strong responses to increasing discharge, which could indicate that, as a result of weathering, Si can accumulate in the weathered zone. Thus, during intensive rainfall, it can be washed out from this zone into surface water [30–32]. Moreover, biologically associated solutes such as DOC (dissolved organic carbon), TOC (total organic carbon), NH$_4$$^+$, and PO$_4$$^{3-}$ often yield positive slopes and show flushing behaviors [17,33], suggesting that the content variability of these biological solutes is minor relative to discharge. It was reported that contents of biological solutes are often linked to anthropogenic sources, influencing these solute flushing behaviors in forested and agricultural catchments, especially in extreme climate events [17,34]. Our results from plots of *b* versus *CV*$_C$/*CV*$_Q$ < 0.5 are consistent with the general interpretation of biogeochemical source presented in the previous study catchment [12,17].

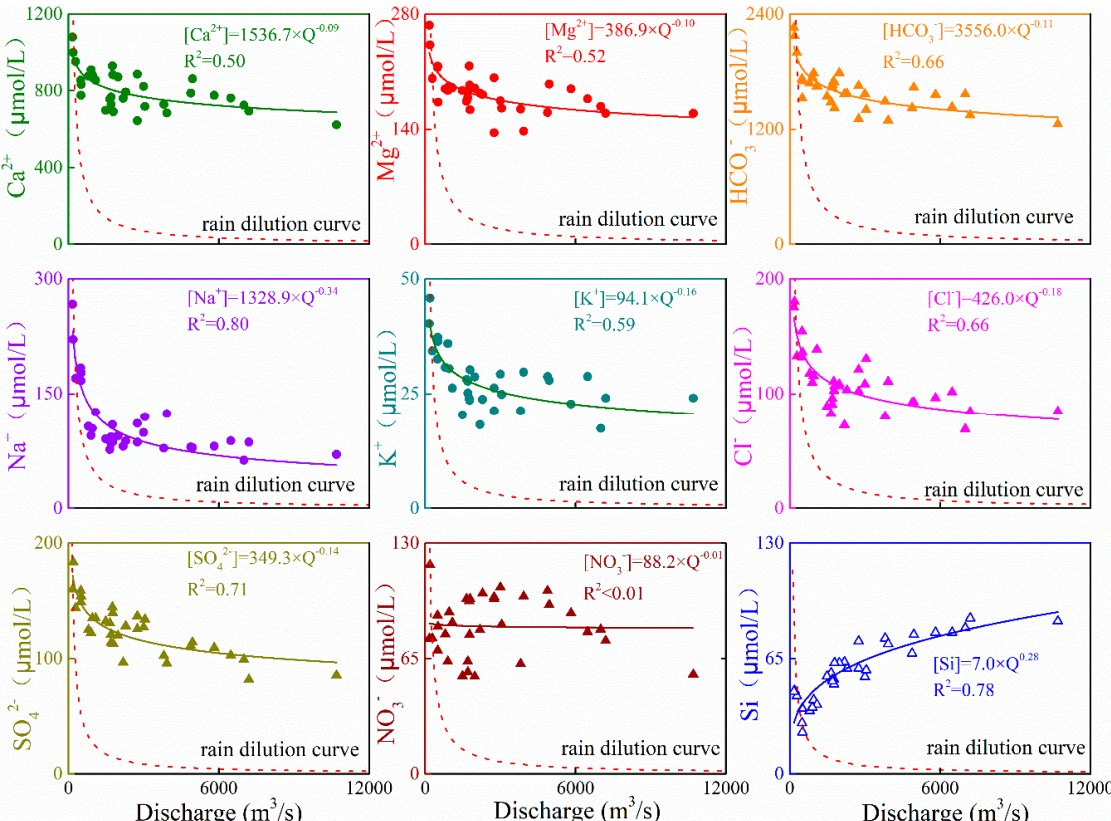

**Figure 2.** Riverine dissolved solutes content–discharge relationships in the Liujiang River.

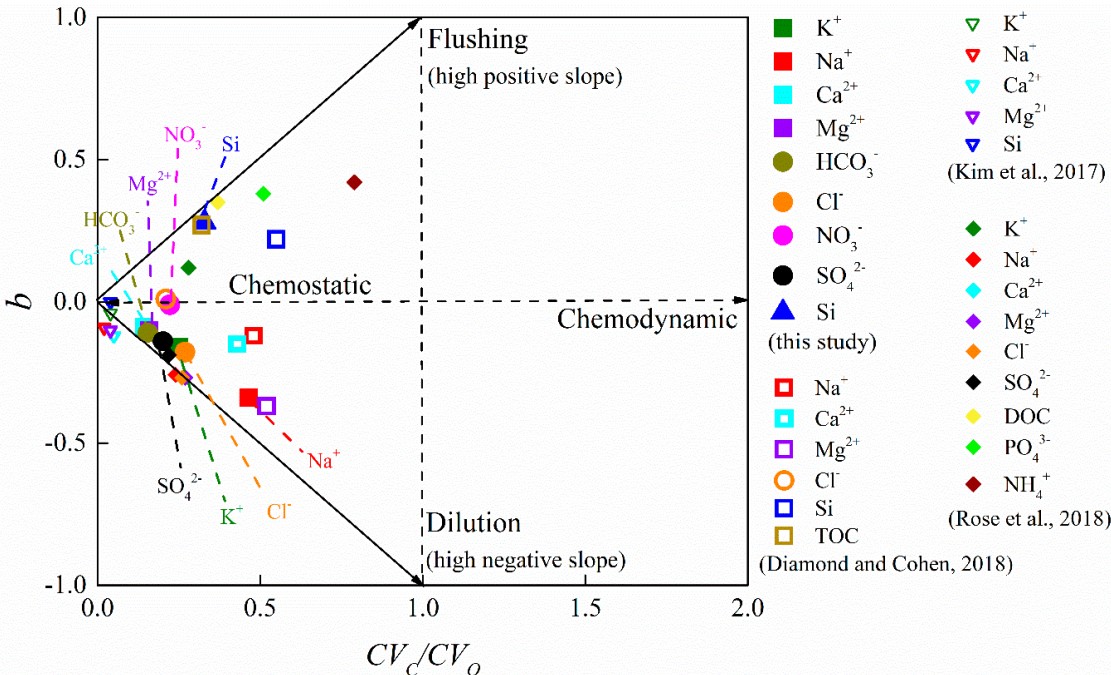

**Figure 3.** Plot of slope (*b*) versus $CV_C/CV_Q$ for riverine dissolved solutes in the Liujiang River.

*4.2. Response of Sulfur Source to Hydrological Variations*

Riverine sulfate is generally derived from atmospheric precipitation, sulfide oxidation, evaporite dissolution, and anthropogenic inputs (such as coal mining and combustion). The chemical composition of river water is insufficient to distinguish dissolved $SO_4^{2-}$ from geologic and exogenous sources, but the combined values of $\delta^{34}S_{SO4}$ and $\delta^{18}O_{SO4}$ could potentially provide useful constraints to identify the sulfur sources [3,35,36].

Isotopic signatures of sulfur and oxygen⁻ are well preserved during the congruent dissolution of evaporites, yielding results of $\delta^{34}S_{SO4}$ varying from 13‰ to 15‰ and $\delta^{18}O_{SO4}$ from 14.5‰ to 32.5‰ [37,38]. The $\delta^{34}S_{SO4}$ values exceed 20‰ along with high contents of $Ca^{2+}$ and $SO_4^{2-}$ in the shallow groundwater of the North Chinese Plain, which are associated with the evaporites (such as gypsum) dissolution [39]. The oxidative weathering of sulfide, which provides riverine sulfate, usually produces a more negative $\delta^{34}S_{SO4}$ value; meanwhile, typical $\delta^{18}O_{SO4}$ values of reduced sulfur oxidation vary from −5‰ to 4‰ [40], which is consistent with the values of $\delta^{34}S_{SO4}$ and $\delta^{18}O_{SO4}$ in the studied river water samples. In the Wujiang River, the oxidation of sulfides (averaging 73%) contributes most of the riverine sulfate [3]. Additionally, Turchyn et al. [35] reported that the headwaters of the Marsyandi River exhibit light $\delta^{34}S_{SO4}$ and $\delta^{18}O_{SO4}$, which can be attributed to the anoxic weathering of pyrite via $Fe^{3+}$. Southern China is also one of the regions that is most affected by acid rain contributed by high sulfur-content coal combustion. Li et al. [41] proved that the source of dissolved riverine $SO_4^{2-}$ in the Jialing River is likely to be due to high S content coal combustion and oxidation of sulfides during the weathering of coal-containing strata. The $\delta^{34}S_{SO4}$ values of rainwater in Guiyang City are reported to have an average of 4.6 ± 5.0‰ [42]. The $\delta^{18}O_{SO4}$ values of atmospheric precipitation are reported to be from 7‰ to 17‰ [40]. The $\delta^{18}O_{SO4}$ of the Liujiang River averages 6.8‰, which shows the dominant contribution of atmospheric input to sulfate in the river water. As indicated by diagrams of $\delta^{34}S_{SO4}$ versus $\delta^{18}O_{SO4}$ in the Liujiang River (Figure 4), riverine sulfate is mainly sourced from three major sources, including atmospheric precipitation, sulfide oxidation, and evaporites. Although a potential contribution from anthropogenic input can affect the sulfur levels [43], previous related works demonstrated that anthropogenic input only contributes a small fraction in the chemical weathering processes in the study area [23,24,44]; hence, the anthropogenic input to riverine sulfate can be ignored.

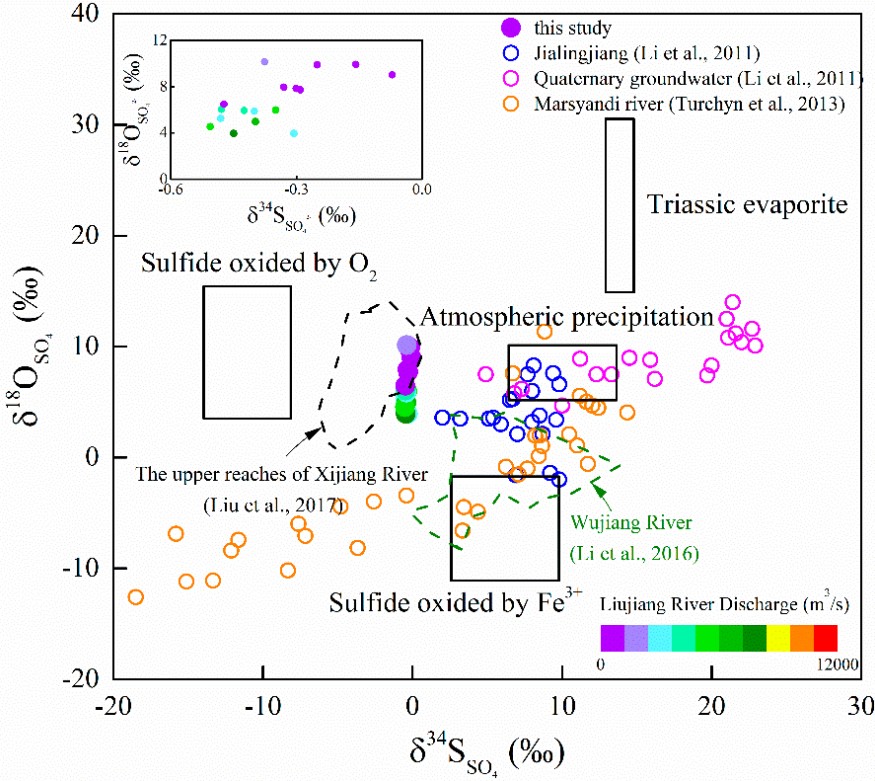

**Figure 4.** A diagram of $\delta^{34}S_{SO4}$ versus $\delta^{18}O_{SO4}$ showing the riverine sulfate sources produce isotopically distinct sulfate in the Liujiang River [3,35,39,41,45].

According to the discussion above regarding the sulfur sources in the Liujiang River, for the $\delta^{34}S_{SO4}$ of the river water, a balanced equation could be calculated based on the following equation:

$$\delta^{34}S_{SO4\ riv} = F_{at} \times \delta^{34}S_{SO4\ at} + F_{sul} \times \delta^{34}S_{SO4\ sul} + F_{gyp} \times \delta^{34}S_{SO4\ gyp} \tag{1}$$

$$F_{at} + F_{sul} + F_{gyp} = 1 \tag{2}$$

where $F_{at}$, $F_{sul}$, and $F_{gyp}$ are the corresponding relative contributions of $SO_4{}^{2-}$ from atmospheric precipitation, oxidation of sulfide, and gypsum input. The end-members can be assigned as follows: $\delta^{34}S_{SO4\ at} = 1.4‰$, $\delta^{34}S_{SO4\ sul} = -13‰$, and $\delta^{34}S_{SO4\ gyp} = 25.7‰$. Specific analysis can be found in related studies [46]. The fractions of $SO_4{}^{2-}$ contributed by the three end-members to the Liujiang River were estimated by using the IsoSource (v1.3, http://www.epa.gov/wed/pages/models) program with an increase of 1.0% and a mass balance tolerance of 0.5%. From Figure 5, it can be seen that atmospheric precipitation (averaging 52%) and the oxidation of sulfide (averaging 35%) are the major sources of $SO_4{}^{2-}$ in the study river, followed by gypsum (averaging 13%). The values of $F_{sul}$ and $F_{gyp}$ contribute to riverine sulfate and show negative relationships with increasing discharge, indicating that these geological sulfate sources exhibit strong dilution behavior with respect to discharge change, while the $F_{at}$ values show positive relationships with discharge changes. Such variations in hydrological connectivity can be attributed to the geogenic sulfate-rich groundwater predominant in the low flow season and greater contributions from an exogenous source, such as low content of atmospheric precipitation and soil water in the high flow season, which are the main drivers of the chemostatic behavior of total riverine sulfate responding to increasing discharge [17]. Therefore, these results suggest that the specific dual isotopic characteristics of riverine sulfate not only reflect the mixing of compositionally distinct end-members but play an important role in better understanding the hydrological variability of riverine sulfate sources in the studied watershed across multiple events and years.

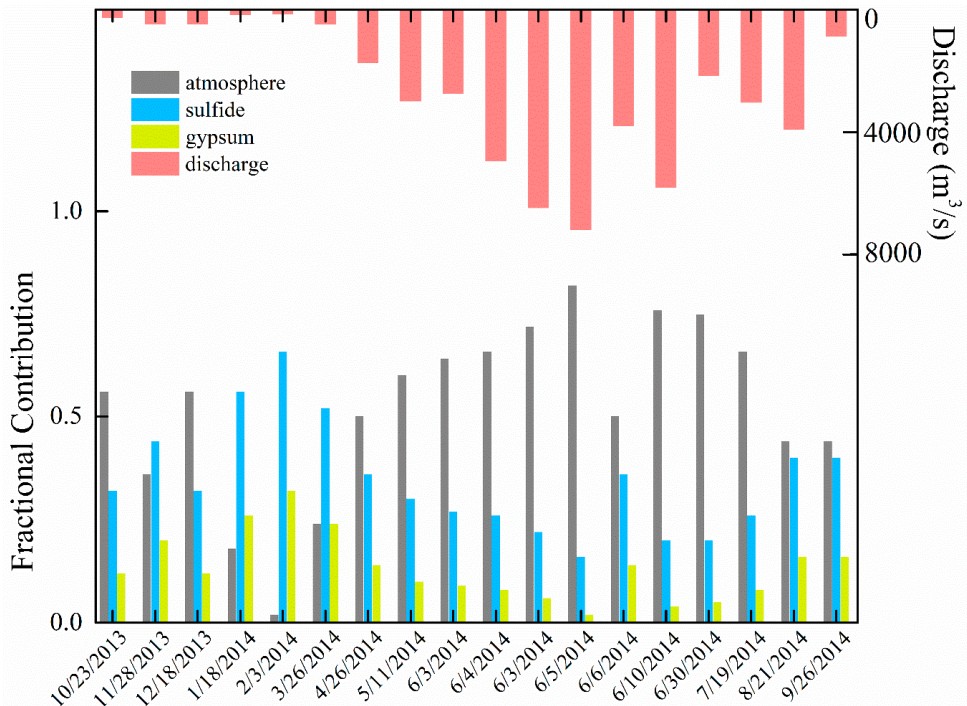

**Figure 5.** The fractions of riverine sulfate contributed by different three end-members ($F_{at}$, $F_{sul}$, and $F_{gyp}$) to the Liujiang River.

### 4.3. Chemical Weathering and $CO_2$ Consumption Are Affected by Climate Variability

Along with the comparison of similar geological backgrounds (predominantly carbonates) from world rivers, good relationships are observed between $Ca^{2+}/Na^+$ vs. $HCO_3^-/Na^+$ (Figure 6a) during a hydrological year in the study river. All river waters fall on a line showing the mixing between carbonate and silicate end-members. River water samples contain more carbonate signature in the high flow conditions than those in the low flow conditions, which can be attributed to, as previously discussed, the stronger chemostatic behavior of carbonates weathering (indicated by $Ca^{2+}$) than silicates weathering (indicated by $Na^+$). As Figure 6b shows, the water chemistry of the studied river is attributed to carbonate weathering involving not carbonic acid but sulfuric acid, which can be mainly sourced from atmospheric precipitation and sulfide oxidation, as analyzed above (see Section 4.2). For the investigated river, it is worth noting that $Ca^{2+}/Na^+$, $HCO_3^-/Na^+$, and $SO_4^{2-}/Na^+$ molar ratios of river water samples are also positively correlated with temperature, suggesting that, in addition to discharge, the temperature may be another variable that is usually and highly correlated with chemical weathering, implying that there may be a negative feedback between climate conditions and chemical weathering that could help in regulating atmospheric $CO_2$ [2,28,47].

Based on the mass budget equations, a forward model is employed to quantify the relative contribution of different sources to the dissolved ions in the river [4,24], which were reported in the related studies [45]. In total, the dissolved ions are dominated by carbonate weathering, accounting for a 61% average for the investigated river, which is followed by the average contribution of atmospheric precipitation (17%), anthropogenic input (8%), sulfide oxidation (7%), gypsums (4%), and silicate weathering (2%) in the Liujiang River (Figure 7). For anthropogenic input, the contribution percentage generally varies from 6% to 11%. There is an increasing proportional contribution from atmospheric precipitation responding to increasing discharge, suggesting the dilution effect. The proportion from sulfide oxidation and gypsums responding to discharge changes is constant with the behavior of their sulfate geogenic sources in the study river (see above Section 4.2). The proportion from the contribution of carbonate weathering not only shows a chemostatic behavior in respect of discharge changes but varies with a similar tendency as the water temperature change. These can be attributed to,

under high temperature and discharge conditions, hydrological flushing of subsurface materials that could further induce the water–rock interaction [28], thus leading to the high intensity of carbonate weathering by reacting with soil $CO_2$ and plant biological processes in the warm-wet environment [25]. Hence, relative to silicate weathering, carbonate weathering shows a stronger chemostatic behavior to respond with increasing discharge, while silicate weathering has more sensitivity to respond with increasing discharge, which is in agreement with the observations reported in climatic-impacted world rivers [9,12,28,48].

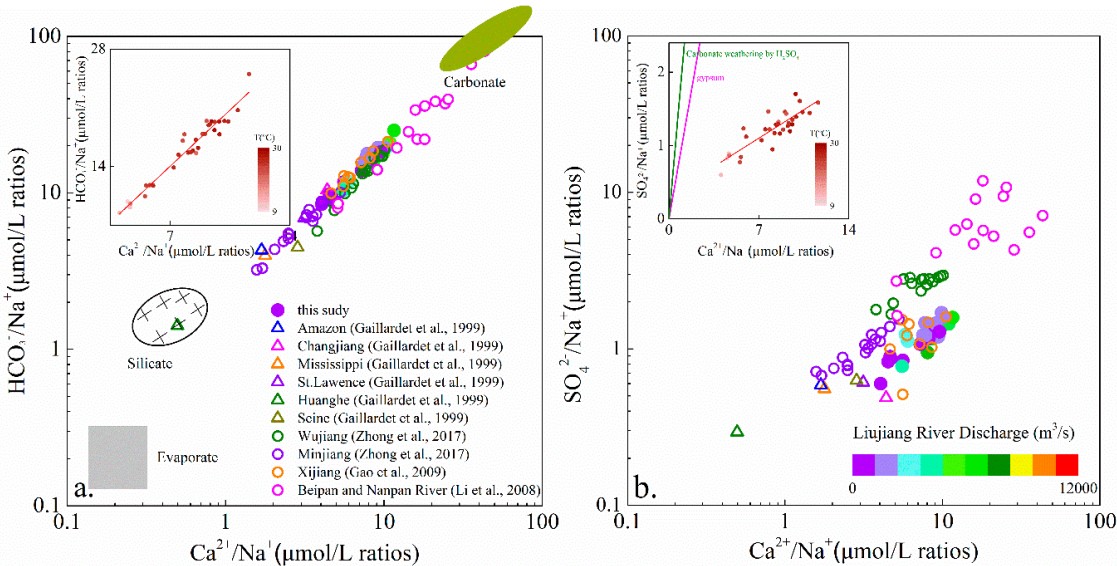

**Figure 6.** Plots of (a) $Ca^{2+}/Na^+$ versus $HCO_3^-/Na^+$ and (b) $Ca^{2+}/Na^+$ versus $SO_4^{2-}/Na^+$ for the Liujiang River waters in a hydrological year [1,4,9,11,23].

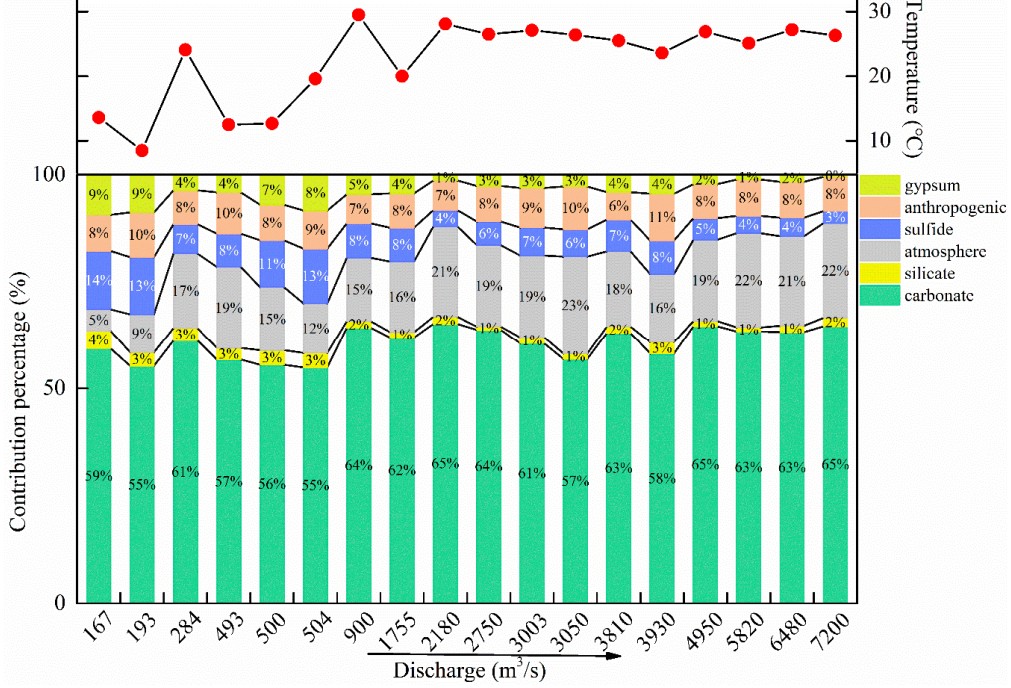

**Figure 7.** The contribution percentage of different end-members under various temperature and discharge conditions for the Liujiang River in a hydrological year.

The $CO_2$ consumption fluxes of silicate weathering ($FCO_{2sil}$) and carbonate weathering ($FCO_{2carb}$) are deducted from the sulfuric acid consumed by silicate and carbonate weathering, respectively. The total $CO_2$ consumption flux ($FCO_2 = FCO_{2sil} + FCO_{2carb}$) in the Liujiang River ranges from 7.7 kg/day/km$^2$ to 237.8 kg/day/km$^2$, averaging 95.4 kg/day/km$^2$, which is slightly higher than the values from Xu and Liu [24]. Meanwhile, both discharge and temperature are positively correlated with chemical weathering, as indicated in Figure 8. In the low flow season, when discharge is < 2000 m$^3$/s while temperature covers a wide range of change (8.5–29.5 °C), $FCO_2$ is more sensitive to variations in temperature than discharge; on the other hand, in the high flow season with high temperature, the discharge is likely to be a dominant driver in transporting the chemical weathering materials and stimulating the chemical weathering intensity by enhancing the available water–rock reaction surface area [2]. The observed positive relation between $CO_2$ consumption and climatic conditions in the study river is consistent with the Loch Vale River [2], the Xijiang River [12], and other relative studies that used a spatial approach to investigate impacts on $CO_2$ consumption flux by chemical weathering [4,5]. The results from this study have important implications in light of regulations for $CO_2$ in the air; if discharge and temperature increase in response to climate change, $CO_2$ consumption flux by chemical weathering may respond similarly, providing a negative feedback on greenhouse $CO_2$ content.

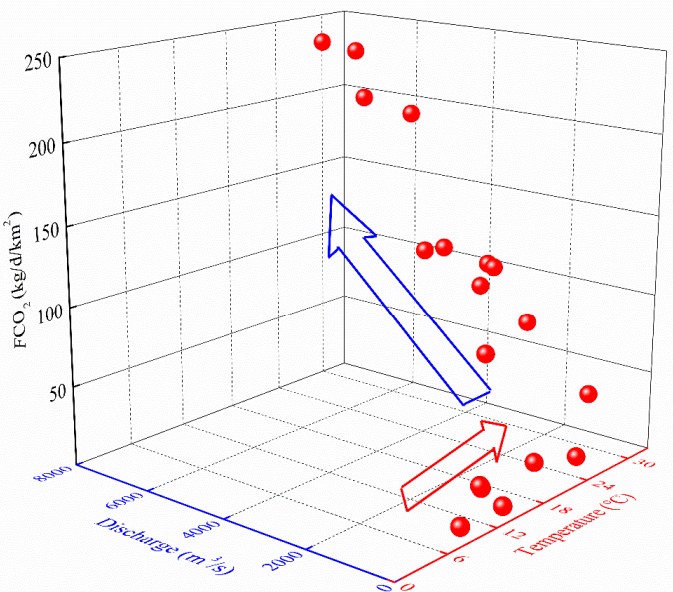

**Figure 8.** Three-dimensional representation of $FCO_2$, discharge, and temperature for the Liujiang River in a hydrological year.

### 4.4. Riverine Carbon Dynamic in Hydrological Variation

Dissolved inorganic carbon represents the largest fraction of the annual fluvial carbon flux to ocean, more than 80% in the Xijiang River [12]. Riverine DIC is mainly sourced from soil $CO_2$, carbonate dissolution, and atmospheric $CO_2$. Due to the high $pCO_2$ in the study river, the contribution of atmospheric $CO_2$ is not considered. Soil $CO_2$, in situ biodegradation, and photosynthesis are the primary drivers of the $pCO_2$ in river water [25,49,50]. Due to the relatively low DOC content and the few aquatic plants in the study catchment, the contributions of biodegradation and photosynthesis to $pCO_2$ could be ignored. Thus, soil $CO_2$ should be a dominant control on $pCO_2$ content. As shown in Figure 9a, the $pCO_2$ contents yield a power-law dilution effect and chemostatic behavior in responding to discharge variation. The dilution signals for $pCO_2$ contents could result from contributions of high-content soil $CO_2$ in the low flow conditions that become increasingly exhausted in the high flow conditions, while the chemostatic behavior for $pCO_2$ contents with respect to increasing discharge

should be attributed to exogenous soil $CO_2$ discharged into river water, especially in extreme climatic events such as storms, when the exogenous soil $CO_2$ counteracts the dilution effects [12,25].

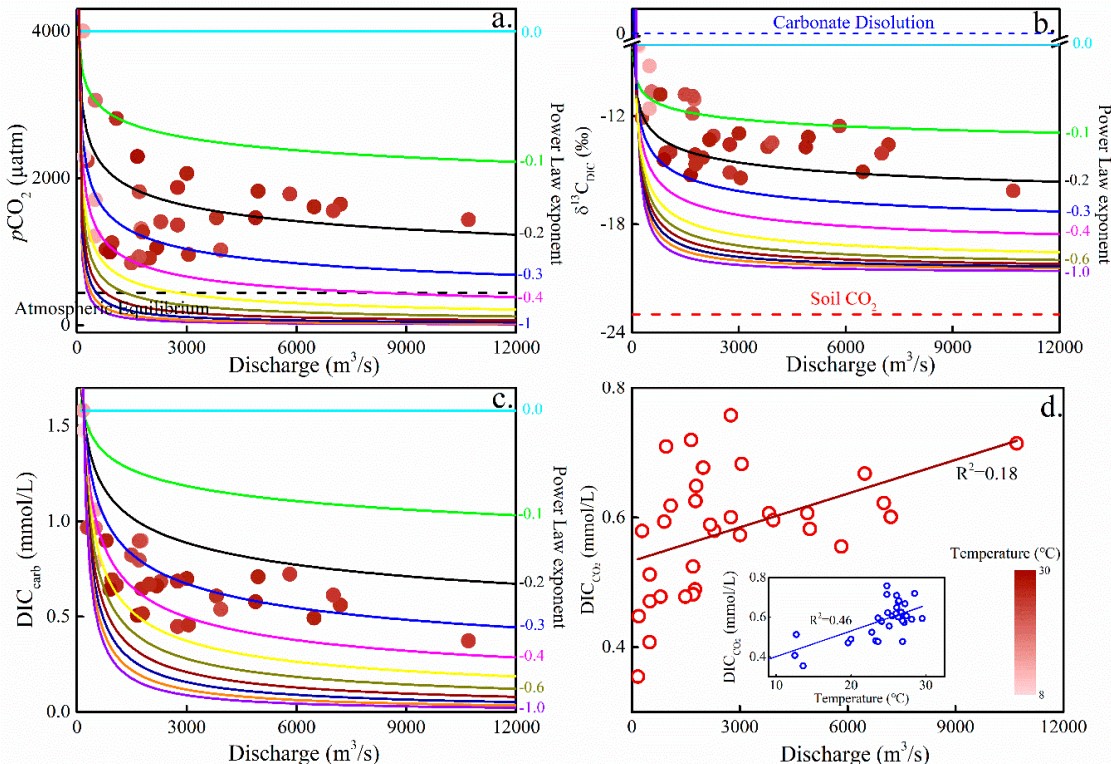

**Figure 9.** The power–law relationship between $pCO_2$ and discharge (**a**). $\delta^{13}C_{DIC}$ values responding with discharge variations (**b**). The power–law relationship between $DIC_{carb}$ and discharge (**c**). The power–law relationship between $DIC_{CO_2}$ and discharge (**d**).

The soil $CO_2$ decreases $\delta^{13}C_{DIC}$ values in the study river, as indicated in Figure 9b, with $\delta^{13}C_{DIC}$ values showing a similar behavior to $pCO_2$ contents responding to hydrological variations. Therefore, $\delta^{13}C_{DIC}$ can be used to constrain the riverine DIC sources. The major vegetation in the study catchment is $C_3$ plants with a mean $\delta^{13}C_{DIC}$ value of −27‰ [1]. After considering isotopic fractionation 4.4‰ [51], the $\delta^{13}C_{DIC}$ of soil water should be −22.6‰. Carbonate carbon has a mean value of 0‰. Therefore, the proportions of two DIC sources are calculated as follows:

$$\delta^{13}C_{DIC\ riv} = F_{CO_2} \times \delta^{13}C_{CO_2} + (1\text{-}F_{CO_2}) \times \delta^{13}C_{Carb} \tag{3}$$

where $F_{CO_2}$ is the proportion of soil $CO_2$, and $\delta^{13}C_{DIC\ riv}$, $\delta^{13}C_{CO_2}$, and $\delta^{13}C_{Carb}$ are $\delta^{13}C$ values of river, soil $CO_2$, and carbonate dissolution, respectively. Based on the mixing model, 29%–65% (averaging 43%) of the DIC is sourced from carbonate dissolution, while 35%-71% (averaging 57%) of the DIC is from soil $CO_2$. As presented in Figure 9c,d, the contributions of carbonate dissolution to riverine DIC show chemostatic behavior responding to increasing discharge, while the contribution of soil $CO_2$ presents a linear relationship with increasing discharge, implying that soil $CO_2$ is a dominant driver of the chemostatic behavior of total riverine DIC with increasing discharge, which is in agreement with previous studies [25]. Therefore, under high temperature and discharge conditions, rainwater infiltrates into the soil and flushes excessive biological solutes, including soil $CO_2$ into the river, leading to increasing the amounts of soil $CO_2$ to total DIC in the high flow season compared to those in the low flow season.

Our calculation indicates that fluvial DIC content and its carbon isotope primarily reflect the mixing of compositionally distinct riverine DIC sources and biogeochemical processes in response to

hydrological changes. These results are important, as they clearly show that physical and biological processes affect the DIC pool with respect to hydrological variations, and the $\delta^{13}C_{DIC}$ can be used to constrain carbon evolution. The riverine dissolved carbon dynamic in response to hydrological variations can be considered a positive feedback in the geological and the biological carbon cycle and a negative feedback in the acidification of ocean by the absorption of atmospheric $CO_2$. Therefore, investigations of long-term carbon dynamics within a single catchment and across multiple catchments incorporating data from multiple storm events over many years should be campaigned.

## 5. Conclusions

According to a high-frequency variation in riverine solutes contents and multiple sable isotopic tracers (carbon and sulfur isotopes), this study investigated chemical weathering, $CO_2$ consumption, and riverine solute sources and their contributions impacted by climatic variabilities in the typical monsoonal river. The variability of solute content is generally much smaller than that of climatic variability, which would support a similar chemostatic behavior. The main reason can be attributed to carbonates dissolution and biological processes. In this study catchment, carbonate weathering controls the major solute source and shows strong chemostatic behavior owing to the rapid dissolution characteristics. On the other hand, along with high temperature, primary production is increased in the high flow season, leading to the influx of $\delta^{13}C$-depleted soil $CO_2$ being the main driver controlling the riverine DIC dynamics. Moreover, the positive correlations between $CO_2$ consumption fluxes and discharge and temperature provide a negative feedback on the greenhouse $CO_2$ in the atmosphere. Quantifying the strength of the feedback between $CO_2$ consumption fluxes and climate change in a range of catchments needs to be addressed in future studies.

**Supplementary Materials:** The following are available online at http://www.mdpi.com/2073-4441/12/3/862/s1, Table S1: The hydrochemistry and stable isotopes for the Liujiang River.

**Author Contributions:** Conceptualization, J.L., H.D. and M.X.; Formal analysis, H.D. and M.X.; Funding acquisition, J.L.; Investigation, M.X., Z.-Y.X., Y.W., J.-T.P., H.W. and X.-D.W.; Methodology, L.Z.; Software, Z.-H.S.; Writing—original draft, J.L. All authors have read and agreed to the published version of the manuscript.

**Funding:** This work was supported financially by National Natural Science Foundation of China (Grant Nos. 41803022, 41964005), the Guizhou Science and Technology Department Fund (Grant Nos. [2013]3118, [2019]1043), Guizhou Education Department Fund (Grant Nos. [2018]139, [2018]161) and scientific platform talent project of Guizhou University of Finance and Economics (Grant No. [2018]5774-029).

**Conflicts of Interest:** The authors declare no conflict of interest.

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
