# Peer review of "Climatic Variabilities Control the Solute Dynamics of Monsoon Karstic River: Approaches from C-Q Relationship, Isotopes, and Model Analysis in the Liujiang River"

_water, doi:10.3390/w12030862_

Round 1
Reviewer 1 Report
Overall this is an interesting paper on an important topic, namely climatic variabilities control the solute dynamics of monsoon karstic river: approaches from C-Q relationship, isotopes and model analysis in the Liujiang River. This is a well-written manuscript, with clear and useful figures. The dataset is very interesting and results are quite clear. The paper presents some useful and interesting case study data, and the overall results and inferences being presented are sound, and of wider interest.
Main points:
1. Care may be needs to be taken in making direct inference of the direct link between solute content response and discharge (in 4.1 section), when the data being compared are only instantly chemistry sampling against daily recharge mean.
2. Lines 172-174 – unclear wording in this sentence, suggest re-phrasing the explanation of how Si shows strong responses to increasing discharge. Is it related to flushing behaviors ?
Which could be the sources of Si in study area?
3. In figure 5 values of the expected uncertainty of the mixture fraction should be given.
Reviewer 2 Report
Nicely written manuscript with a good presentation and evaluation of data.
If i really go in the very tiny details the only question that I would have for the authors is to extend the sampling and analysis section as there they were referring to the literature and not explaining the method in detail. Same would be for the modelling part.
Data are presented and explained nicely. For the first time I also see that the literature was selected properly without favoring one or another research group.
Reviewer 3 Report
- Authors highlight the impact of climatic variability on solutes dynamics and chemical weathering. Therefore, it is recommended to show in the article the characteristics of precipitation and air temperatures during the research period. This will help the reader to assess the influence of climatic factors on the concentration-discharge relationship (C-Q).
- The authors' research shows that the C-Q relationship for silicon is described by the value b>0. The research conducted, among others, by WELS & al. 1991, ASANO & al. 2003, DOBRZYŃSKI 2005, showed that as a result of weathering, silicon can accumulate in the weathered zone. During intensive rainfall it can be washed out from this zone into surface water. Therefore, the flushing behaviors of Si are similar to the behaviour of biologically related solutes such as DOC, TOC, and others. It should be clearly emphasized in the article that the concentration of Si is not mainly dependent on biological processes. The phrases used by the authors, for example „whereas exogenous solutes (like atmospheric input to riverine sulfate) and biological solutes (like Si and soil CO2)”, may mislead the reader (See v. 22 page 1 and v.172-174 page 4-5).
- v. 106-107 page 3: In section 2.2, please provide information whether samples for the determination of ICP-OES were filtrated through 0.45 µM cellulose-acetate membrane paper.
- v. 110 page 3: In section 2.2, please specify which standard was used for the determination of δ13CDIC
- Figure 4 and Figure 6: The headline "Discharge" should be changed to "Liujiang River Discharge". It's not obvious that the colored scale refers to Liujiang River Discharge.
- v. 254 page 8 and v. 271 page 9: Please correct the quotation - "see above 5.2"
- v. 262 page 9: It should be pointed out in the text that the forward model used is based on mass budget equations.
- v. 110 page 3 and v. 203 page 6: Please check the literature references
ASANO, Y., UCHIDA, T. & OHTE, N. 2003. Hydrologic and geochemical influences on the dissolved silica concentration in natural water in a steep headwater catchment. Geochimica et Cosmochimica Acta, 67 (11), 1973-1989.
WELS, C., CORNETT, R.J. & LAZERTE, B.D. 1991. Hydrograph separation: a comparison of geochemical and isotopic tracers. Journal of Hydrology, 122, 253-274.
DOBRZYŃSKI, D. 2005. Silica origin and solubility in groundwater from the weathered zone of sedimentary rocks of the Intra-Sudetic Basin, SW Poland. Acta Geologica Polonica, 55 (4), 445-462. Warszawa.
